# Do family and maternal background matter? A multilevel approach to modelling mental health status of Australian youth using longitudinal data

**Rubayyat Hashmi**[1,2]*, **Khorshed Alam**[2], **Jeff Gow**[2,3], **Sonja March**[4]

**1** Department of Economics, American International University-Bangladesh (AIUB), Dhaka, Bangladesh, **2** School of Business, Faculty of Business, Education, Law & Arts, University of Southern Queensland, Toowoomba, Queensland, Australia, **3** School of Accounting, Economics and Finance, University of KwaZulu-Natal, Durban, South Africa, **4** School of Psychology and Counselling, Faculty of Health, Engineering and Sciences University of Southern Queensland, Springfield, Queensland, Australia

☉ These authors contributed equally to this work.

* rubayyat@gmail.com

**Data Availability Statement:** Melbourne Institute of Applied Economic and Social Research (MIAESR) is the data custodian of the HILDA survey. Access to this data is restricted, and it is

## Abstract

### Purpose

Most previous research place great importance on the influence of family and maternal background on child and adolescents' mental health. However, age of onset studies indicates that the majority of the mental health disease prevalence occurs during the youth years. This study investigates the relationship of family and maternal background, as well as individual circumstance on youth mental health status.

### Method

Data from 975 participants and 4632 observations of aged cohort 15 to 19 years in the Household, Income and Labour Dynamics in Australia (HILDA) longitudinal study were followed for 10 years (2007–2017). Multilevel logistic regression models were used to analyse the impact of youth circumstances on mental health status.

### Results

The findings suggests that not all dimensions of family and maternal background (especially maternal education) have impacts on youth mental health. We found low household income (AOR: 1.572, 95% CI: 1.017–2.43) and adverse living arrangement (AOR: 1.586, 95% CI: 1.097–2.294) significantly increases mental disorder odds whereas maternal education or occupation fixed effects were not significant. Individual level circumstances have much stronger impact on youth mental health. We found financial shock (AOR: 1.412, 95% CI: 1.277–1.561), life event shock (AOR: 1.157, 95% CI: 1.01–1.326), long term health conditions (AOR: 2.855, 95% CI: 2.042–3.99), smoking (AOR: 1.676, 95% CI: 1.162–2.416), drinking (AOR: 1.649, 95% CI: 1.286–2.114) and being female (AOR: 2.021, 95% CI: 1.431–2.851) have significant deteriorating effects on youth mental health.

not publicly available. The data is available from the National Centre for Longitudinal Data (NCLD) of Department of Social Services (DSS) for researchers of approved organisations who meet the criteria for access to confidential data. Those interested in accessing this data should contact the NCLD and Australian Data Archive (ADA). Those who apply to access the data need to complete the confidentiality deed poll and send the copy to NCLD (ncldresearch@dss.gov.au) and ADA (ada@anu.edu.au). The following web link has the guideline for the application: https://dataverse.ada. edu.au/dataverse.xhtml. To contact the national office: 71 Athllon Drive, Greenway ACT 2900, GPO Box 9820, Canberra ACT 2601. DSS national number: 1300 653 227, For International Callers: + 61 2 6146 0001. To contact MIAESR: Melbourne Institute of Applied Economic and Social Research, The University of Melbourne, VIC 3010, Australia (Tel: +61 3 8344 2073 or Fax: +61 3 9347 6739).

**Funding:** The authors received no specific funding for this work.

**Competing interests:** The authors have declared that no competing interests exist.

## Conclusions

Our finding is in contrast to the majority of studies in the literature which give a preeminent role to maternal characteristics in child and youth mental health status. Mental health interventions should consider heterogeneity of adverse youth circumstances and health-related behaviours.

## Introduction

Social gradients in physical and mental health status exist and social justice demand that mental health inequality is minimised [1–3]. Thus, understanding the determinants of socioeconomic inequality is important for policy makers and researchers alike. While socioeconomic inequalities in adult mental health dominates current research, a growing body of literature currently also suggests that lower socioeconomic status (SES) is an important determinant of mental health problems in children and adolescents [4–6]. Furthermore, since childhood and adolescence are the critical ages for the onset of mental illnesses, mounting evidence also suggests that maternal background plays an important role in the social determinants of children and adolescents' mental health [7–9].

Although child and adolescent periods appear to be emerging points for mental disorders, age of onset (AOO) studies have identified that the majority of mental disorder incidence occurs at the early stages of youth, particularly when young people transition to adulthood [10, 11]. The problem in the existing literature is that the age bands in these studies are broad, obscuring the stages of youth by either younger youths being included with 'children and adolescents' (e.g. age 1–18 years) or older youths being included with 'adults' (e.g. 15–64 years) [5, 12, 13]. The circumstances (e.g., life chances, opportunities and adverse events) experienced by individuals in their childhood and adolescent period are certainly much different than the period when they are transitioning to youth and adulthood. Thus, the impact of family and maternal background on this transitioning phase (15–19 years) on an individual's mental health outcome is not clear and may very well be different.

In this paper, we tried to address this issue by selecting a 15–19 years age cohort and following the cohort for ten years (up to six measurement points) to investigate the impact of youth circumstances (adverse life events experiences, family and maternal background characteristics, household characteristics etc.) on mental health outcomes. Although significant advances have been made in our understanding of the impact of family and maternal background on childhood mental health status, considerable knowledge gaps still exist. For instance, how different attributes that constitutes family and maternal social class variations (such as mothers' education, income or occupational status) contributes to the variation in youth mental status or how such mental health inequalities evolve over time are not well understood in the literature [14, 15]. Little is known about the variability of individual level and social class level characteristics on mental health outcome inequalities for youth and young people. In summary, we answer two research questions here i) Which individual and background characteristics of youth circumstances impact youth mental health outcomes? And how much? and ii) Whether family and maternal background contribute to substantial variation in youth mental health status.

Thus, the primary goal of this paper is to improve this knowledge gap and attempt to provide a link between prior studies on childhood and adult mental health inequalities. In addition, the focus on Australian youth complements existing US, UK or European studies on

youth mental health inequalities. Our study extends the literature to another developed country with different social welfare system and norms that provide different perspectives on mental health equity issues.

## Methods

### Data source

All our analyses are based on sample data from the Household, Income and Labour Dynamics in Australia (HILDA) panel survey [16]. This nationally representative household survey has been carried out annually from 2001 through 2018 (waves 1–18). It interviews and subsequently reinterviews all members aged 15 years and over of the same selected household every year. More than 30,000 individuals (40,000+ enumerated) have participated in the survey over the years and on average 15,000 individuals have been interviewed every year. A 90% wave on wave response rates of HILDA survey are comparable with other large longitudinal surveys like the British Household Panel Study (BHPS) or Panel Study of Income Dynamics (PSID) [17]. Details of HILDA sample design, survey response rates and attrition rates can be found elsewhere [17].

### Inclusion criteria of the samples

For the purpose of this study, we limit the sample to young Australians aged 15–19 years (late adolescent period) at the baseline wave (wave 7) and then followed the participants for 10 years (up to six measurement points) which covers youth (20–24 years) and transition to adulthood phase (25–29 years) in the follow up. We chose to start from wave 7, because HILDA survey did not start to collect Kessler Psychological Distress Scale (K10) scores (our main outcome of interest) in earlier waves and it provides the score subsequently in every odd wave (every two years) thereafter. Thus, we constructed an unbalanced panel data using wave 7, 9, 11, 13, 15 and 17. To be included in the analyses, the participants had to be interviewed in the baseline wave 7 and has to appear in at least one of the follow-up waves. Our final sample contains 975 participants across the six waves with a total of 4,632 observations. The 15–19 age cohort was thus followed up to 25–29 years with an average of 5.18 observations per person. The participant flow into the sample is shown in Fig 1.

### Outcome variable, exposure variables and other co-variates

This study uses the Kessler Psychological Distress Scale (K10) as the measure of mental health outcomes and is the main dependent variable for analyses [18]. In clinical practice, the scale is used to assess the likelihood of having a mental disorder; for example, a person with a score of 10–15 has a low risk of having a mental disorder whereas a person with a score of 20–24 is likely to have a mild mental disorder, a score of 25–30 would indicate a likely moderate mental disorder and a person with a score of 30–50 is likely to have a severe mental disorder [19]. In the analyses, we use a dichotomous K10 variable (where a score of greater than 20 depict the likelihood of a mental disorder) as measures of our dependent variable for mental health performance [20].

Following Roemer's equality of opportunity theory [21, 22] we classify all our exposure variables into two types: i) circumstances category and ii) effort category. The theory of equality of opportunity revolves around the goal of compensating for 'negative' circumstances (such as parental background) on health outcomes while controlling the health inequalities generated by effort category variables (such as lifestyle or health habits) that can be attributed to the behaviour of an individual. We use the biological mothers' education level and occupational

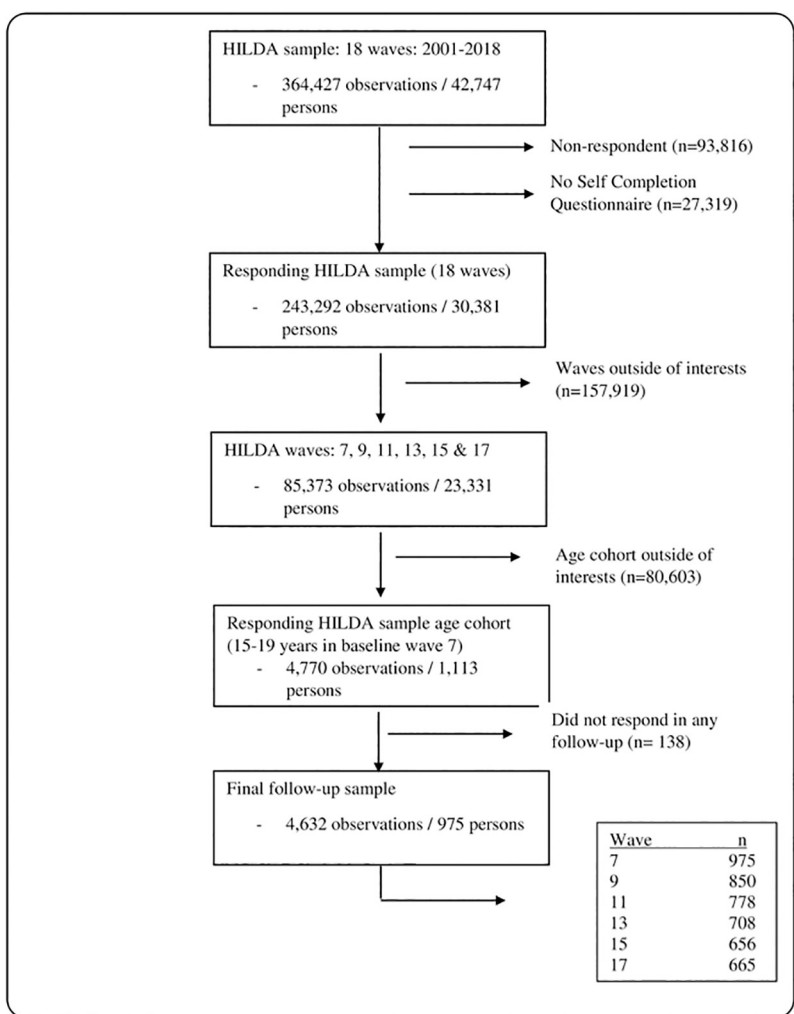

**Fig 1. Participants flow into the sample.**

status, household income and family living arrangements (whether the participant lived with both parents at the age of 14 years old) to determine the family and maternal background status as a group level characteristic of the circumstances category. We define maternal education level as low if the highest qualification level obtained by the mother is secondary level or lower. We use the Australian Socioeconomic Index 2006 (AUSEI06) occupational status scale as the measure of the occupational status of mother [23]. We assign occupational status as low if the value range falls in the lowest quintile. Similarly, we assign household income as low if the equivalised household income range falls in the lowest quintile. Using household income, family living arrangement, maternal education and occupational status we have constructed 16 (2x2x2x2) different types of family and maternal background history groups for the multilevel analyses.

We use the number of financial shocks, number of life event shocks and long-term health conditions in the individual level circumstances category [12]. The number of financial shock variable shows the number of adverse financial events the study participant has experienced (for example: went without meals or asked for financial help from friends or family). Similarly, life event shock variable shows the number of life events related to grief, loss or injury the

study participant has suffered (for example: death of a family member or serious personal injury). The list of events that constitutes financial and life event shocks are given in the S1 Appendix. We use negative health habits such as being obese (as a proxy of unhealthy eating and lack of exercising), being a daily smoker and regular drinker (drinks more than four standard drink/day), and positive health habits such as being an active member of a sporting/ hobby/community-based club or association as an effort type of variables. This study also included gender and rural residency as demographic covariates in the analyses based on past literature [24]. In addition, we construct our time variable by setting zero at the baseline wave 7 and subsequently adding two for each additional measurement point (since between wave time is two years and there are up to six measurement points) to get a ten-year follow-up at wave 17 (t = 0,2,4,6,8, and 10).

## Statistical analyses

The authors constructed an unbalanced longitudinal data set of the youth cohort by linking an individual's record who participated in the baseline (wave 7) at age 15–19 years and in one of the follow-up waves (9, 11, 13, 15 and 17). Descriptive statistics and mental health opportunity profile were summarized to understand the impact of family and maternal background group characteristics on youth mental health. Visual trends of psychological distress scale were analysed for group level characteristics. Traditional single level regression analysis such as logistic regression model only assumes fixed-effect impacts of dependent variables and does not allow for random effects of intercepts and slopes for individual and group level characteristics. However, data structure can be nested or clustered by some observable characteristics that creates similarity between individuals and ignoring these phenomena can violate the independence assumption of regression analysis. Multi-level models allow for a nested data structure (i.e., repeated measures) and make it possible to study sources of variance at different levels of an outcome variable [25]. The nested data structure is illustrated in Fig 2. In our analyses, we used both single level logistic regression and multilevel logistic regression models. we have nested our data structure into three levels: i) time, ii) individual, and iii) family and maternal background history types (a total of 16 different background history types; for example a background history type could be: household income- high; from two types: 'high' and 'low', mothers education- low; from two types: 'high' and 'low', mothers occupation- low; from two types: 'high' and 'low' and family living arrangement—whether not lived with both biological

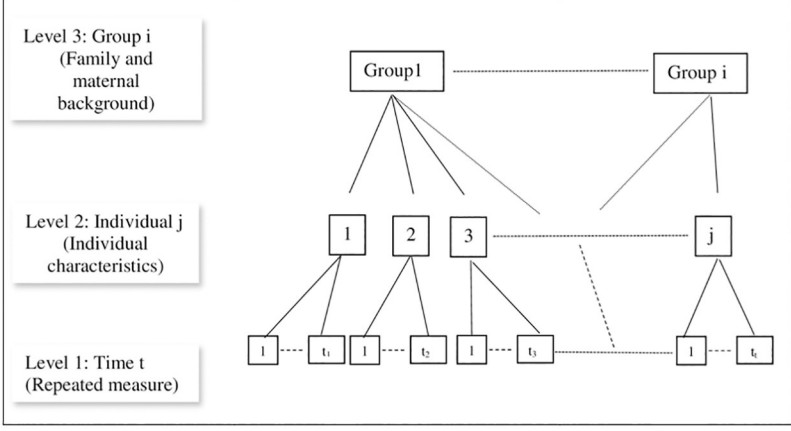

**Fig 2. Repeated measure three level data structure.**

parents- yes; from two types: 'yes' and 'no'. Thus, we have 2x2x2x2 = 16 types. A full combination of 16 types can be seen in Table 2's opportunity profile). We assigned unique identifiers (From 1 to 16, see Table 2's opportunity profile's rank number for identifiers) for each group for the analysis. We control for individual fixed effects characteristics like circumstances and effort covariates in level 2 and group level fixed effects characteristics like various family and maternal background group characteristics in level 3. All statistical analyses were conducted using Stata 15.

## Ethics approval

The HILDA study was approved by the Human Research Ethics Committee of the University of Melbourne. The study used only de-identified existing unit record data from the HILDA survey. The authors completed and signed a confidentiality agreement with NCLD (ncldresearch@dss.gov.au) and obtained database access from the Australian Data Archive (ada@anu.edu.au) following application acceptance. Thus, the dataset studied during this work were subject to the signed confidentiality agreement.

## Results

### Describing the sample

Table 1 displays the socio-demographic characteristics of the study population by mental health status. It can be seen that age groups do not vary substantially in mean K10 score both in the baseline wave and in all waves average. However, in our sample, males have lower average K10 score than females in both baseline wave and all waves average. Richer household

**Table 1. Socio-demographic characteristics of the study population by mental health status.**

|  | Baseline (wave 7) | | All waves | |
|---|---|---|---|---|
|  | N (%) | K10 score Mean (std) | N (%) | K10 score Mean (std) |
| **Gender** | | | | |
| Male | 465 (47.69) | 15.76 (5.87) | 2,109 (45.53) | 16.39 (6.51) |
| Female | 510 (52.31) | 17.78 (6.92) | 2,523 (54.47) | 17.77 (7.25) |
| **Age** | | | | |
| 15 years | 197 (20.21) | 16.62 (6.84) | 197 (4.25) | 16.62 (6.84) |
| 16 years | 240 (24.62) | 16.60 (6.29) | 240 (5.18) | 16.60 (6.29) |
| 17 years | 184 (18.87) | 17.38 (6.43) | 363 (7.84) | 17.22 (6.74) |
| 18 years | 195 (20) | 16.8 (6.26) | 399 (8.61) | 16.84 (6.43) |
| 19 years | 159 (16.31) | 16.77 (6.89) | 466 (10.06) | 16.85 (6.79) |
| **HH Income group (Lowest quintile)** | | | | |
| Low | 222 (22.77) | 18.37 (7.65) | 931 (19.78) | 19.34 (8.32) |
| High | 753 (77.23) | 16.36 (6.07) | 3716 (80.22) | 16.59 (6.46) |
| **Mother's Education (Low = secondary or lower)** | | | | |
| Low | 204 (20.92) | 16.80 (6.67) | 1759 (37.97) | 17.71 (7.27) |
| High | 771 (79.08) | 16.80 (6.48) | 2873 (62.03) | 17.00 (6.87) |
| **Mother's occupational status (Lowest quintile)** | | | | |
| Low | 216 (22.15) | 17.43 (7.12) | 943 (20.36) | 18.46 (7.88) |
| High | 759 (77.85) | 16.64 (6.33) | 3689 (79.64) | 16.80 (6.66) |
| **Did not live with both parents** | | | | |
| No | 652 (66.87) | 16.03 (5.69) | 3169 (68.42) | 16.56 (6.46) |
| Yes | 323 (33.13) | 18.41 (7.7) | 1463 (31.58) | 18.40 (7.79) |

**Table 2. Mental health opportunity profile.**

| Rank | HH income | Mother's education | Mother's occupational status | Did not live with both parents | Group sample size (n) | Average k10 score of the participant | Risk level |
|------|-----------|--------------------|-----------------------------|-------------------------------|----------------------|--------------------------------------|-----------|
| 1 | High | Low | High | No | 328 | 16.1 | Low |
| 2 | High | High | High | No | 2032 | 16.25 | Low |
| 3 | High | Low | Low | Yes | 68 | 16.53 | Low |
| 4 | High | High | Low | No | 208 | 16.62 | Low |
| 5 | High | High | High | Yes | 731 | 17.12 | Low |
| 6 | Low | High | High | No | 231 | 17.17 | Low |
| 7 | Low | High | Low | No | 100 | 17.48 | Low |
| 8 | High | Low | High | Yes | 97 | 17.52 | Low |
| 9 | High | Low | Low | No | 150 | 17.69 | Low |
| 10 | High | High | Low | Yes | 87 | 18.72 | Low |
| 11 | Low | Low | Low | No | 81 | 19.26 | Low |
| 12 | Low | Low | High | Yes | 46 | 19.28 | Low |
| 13 | Low | Low | High | No | 39 | 19.97 | Low |
| 14 | Low | High | High | Yes | 185 | 20.7 | High |
| 15 | Low | Low | Low | Yes | 107 | 20.89 | High |
| 16 | Low | High | Low | Yes | 142 | 21.15 | High |

income group has on average two-point lower K10 scores at baseline and approximately three points in all waves average. Maternal education level does not indicate any significant difference in average scores between education groups. However, mothers with lower occupational status have approximately one point higher average scores. Those youth, who did not live with both biological parents at age 14, have two-point higher average K10 scores both in baseline and all waves average. The standard deviation of the K10 score for the socio-demographic characteristic variables (gender, age, household income groups, mothers' education and occupation, living arrangement) ranges between five and eight points for both the baseline and all waves. This indicates considerable variability of the K10 score at the individual level.

For a deeper understanding of family and maternal background, the mental health opportunity profile of the study participants is provided in Table 2. Depending upon household income, maternal education, maternal occupation and living history arrangement of the participant, 16 types of background groups are identified. The groups are ranked in ascending order according to the average K10 score (lower score implies better mental health). Out of 16 groups, there are three groups with high risk level of developing a mental disorder. Three more groups also show a K10 average of more than 19 and sightly avoid entering into the high-risk group. In addition, the high household income attribute has been clustered into lower rankings and vice versa. To further investigate, we plot the temporal evolution by the 16 family and maternal background types in Fig 3. The thick line (trend values varies between 15 and 25) shows that there also exist a lot of group level variability overtime in the average K10 scores. The trend analysis thus indicates both individual and group level variability and justifies analysing the data through a multi-level modelling approach.

## Regression analysis

The results of the regression models are in Table 3. Since, a single point change in the average K10 score might not mean anything unless it drives up into other risk categories Table 3 considers a dichotomous dependent variable (K10 $\geq$ 20 implies a higher risk of mental disorder) which measures risks through nonlinear estimation of odds ratios. The 'null' model results are

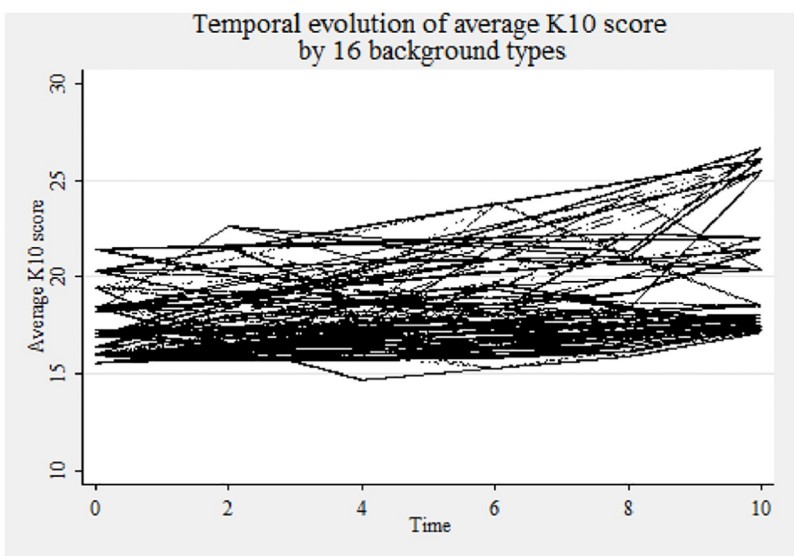

**Fig 3. Temporal evolution of mental health status (K10 score) by background.**

shown in the first column. The 'null' model considers no explanatory variable and focuses just between and within individual variability. The random effect variances estimate for both family and maternal background level (level 3 $\sigma^2_{v0}$ is 0.423 and SE is lower at 0.202) and individual level (level 2 $\sigma^2_{u0}$ is 4.101 and SE is also much lower at 0.422) of the null model justifies the use of the multi-level approach. The second model in Table 3 shows the fixed effect logit estimates for comparison purpose. Unlike multilevel (ML) models, the logit does not have a random component and only shows fixed effects of the variables. To understand the family and maternal background variability, we do not consider the fixed effect of family and maternal background in the third model (Mixed 1 multilevel model). However, the final multilevel model (mixed 2) considers family and maternal background fixed effects. Individual fixed effects are considered in all models.

The individual level circumstances category variables are highly significant in all models. For example, exposure to an additional financial shock has a 1.4 times higher risk of having a mental illness than individuals who do not experience a shock (logit Adjusted Odds Ratio [AOR]: 1.321, 95% CI: 1.243–1.404; Mixed 1 AOR: 1.436, 95% CI: 1.298–1.589 and Mixed 2 AOR: 1.412, 95% CI: 1.277–1.561). Similarly, a single life event shock increases the risk of having mental disorder by 1.15 times higher (logit AOR: 1.156, 95% CI: 1.059–1.262; Mixed 1 AOR: 1.161, 95% CI: 1.013–1.331 and Mixed 2 AOR: 1.157, 95% CI: 1.01–1.326). This is considerable if you consider the possibility of experiencing multiple life events and financial shocks in a period. In addition, the study results also show that individuals who have long term health conditions are approximately 2.9 times highly likely to have a mental condition (logit AOR: 2.232, 95% CI: 1.853–2.688; Mixed 1 AOR: 2.934, 95% CI: 2.098–4.103 and Mixed 2 AOR: 2.855, 95% CI: 2.042–3.99).

The individual effort or lifestyle category variables such as 'daily smoker', 'heavy drinker' and 'active membership of club or sporting activities' are also significant in all models. Club activities have a positive effect on mental health (logit AOR: 0.651, 95% CI: 0.559–0.758; Mixed 1 AOR: 0.623, 95% CI: 0.487–0.797 and Mixed 2 AOR: 0.635, 95% CI: 0.496–0.812). On the contrary, negative habits such as smoking (logit AOR: 1.241, 95% CI: 1.018–1.512; Mixed 1 AOR: 1.801, 95% CI: 1.246–2.604 and Mixed 2 AOR: 1.676, 95% CI:1.162–2.416)

**Table 3. Parameter estimates of different logit regression models.** (Depedent variable: Whether likely to have mental disorder- i.e. K10 $\geq$ 20).

| Fixed effects | Null Estimate (OR) | Std error | 95% CI | Logit Estimate (AOR) | Std error | 95% CI | Mixed 1 Estimate (AOR) | Std error | 95% CI | Mixed 2 Estimate (AOR) | Std error | 95% CI |
|---|---|---|---|---|---|---|---|---|---|---|---|---|
| Intercept | 0.296*** | 0.059 | (0.2–0.437) | 0.168*** | 0.017 | (0.138–0.204) | 0.075*** | 0.018 | (0.047–0.12) | 0.05*** | 0.011 | (0.322–0.078) |
| Wave (time) | | | | 1.019 | 0.011 | (0.998–1.04) | 0.999 | 0.022 | (0.957–1.042) | 1.00 | 0.022 | (0.961–1.047) |
| *Individual characteristics* | | | | | | | | | | | | |
| Gender—Female (Ref.: Male) | | | | 1.484*** | 0.108 | (1.286–1.712) | 2.063*** | 0.363 | (1.461–2.913) | 2.021*** | 0.355 | (1.431–2.851) |
| Rural resident—Yes (Ref.: No) | | | | 0.759* | 0.095 | (0.593–0.97) | 0.89 | 0.195 | (0.579–1.366) | 0.899 | 0.197 | (0.586–1.383) |
| Number of financial shock | | | | 1.321*** | 0.041 | (1.243–1.404) | 1.436*** | 0.074 | (1.298–1.589) | 1.412*** | 0.072 | (1.277–1.561) |
| Number of life event shock | | | | 1.156*** | 0.052 | (1.059–1.262) | 1.161* | 0.081 | (1.013–1.331) | 1.157* | 0.08 | (1.01–1.326) |
| Long term health condition—Yes (Ref.: No) | | | | 2.232*** | 0.212 | (1.853–2.688) | 2.934*** | 0.502 | (2.098–4.103) | 2.855*** | 0.488 | (2.042–3.99) |
| Club activities—Yes (Ref.: No) | | | | 0.651*** | 0.05 | (0.559–0.758) | 0.623*** | 0.078 | (0.487–0.797) | 0.635*** | 0.08 | (0.496–0.812) |
| Daily smoker—Yes (Ref.: No) | | | | 1.241* | 0.125 | (1.018–1.512) | 1.801** | 0.339 | (1.246–2.604) | 1.676** | 0.313 | (1.162–2.416) |
| Heavy drinker—Yes (Ref.: No) | | | | 1.344*** | 0.099 | (1.163–1.554) | 1.651*** | 0.209 | (1.288–2.117) | 1.649*** | 0.209 | (1.286–2.114) |
| Obese—Yes (Ref.: No) | | | | 1.131 | 0.11 | (0.935–1.367) | 1.372 | 0.269 | (0.935–2.014) | 1.311 | 0.256 | (0.895–1.921) |
| *Background characteristics* | | | | | | | | | | | | |
| Household Income—Low (Ref.: High) | | | | 1.258* | 0.116 | (1.05–1.506) | | | | 1.572* | 0.349 | (1.017–2.43) |
| Did not live with both parents—Yes (Ref.: High) | | | | 1.183* | 0.091 | (1.017–1.376) | | | | 1.586* | 0.298 | (1.097–2.294) |
| Mothers Education—Low (Ref.: High) | | | | 0.972 | 0.088 | (0.814–1.162) | | | | 0.921 | 0.203 | (0.597–1.421) |
| Mothers' occupation—Low (Ref.: High) | | | | 1.188 | 0.109 | (0.992–1.423) | | | | 1.314 | 0.296 | (0.845–2.043) |
| **Random effects** | | | | | | | | | | | | |
| Background (level 3) | | | | | | | | | | | | |
| Intercept variance $\sigma^2_{v0}$ | 0.423 | 0.202 | (0.166–1.08) | | | | 0.078 | 0.082 | (0.01–0.608) | 7.14e-32 | 3.89e-17 | |
| Individual (level 2) | | | | | | | | | | | | |
| Intercept variance $\sigma^2_{u0}$ | 4.101 | 0.422 | (3.353–5.017) | | | | 4.068 | 0.718 | (2.878–5.749) | 4.116 | 0.720 | (2.921–5.8) |
| Wave variance $\sigma^2_{u1}$ | | | | | | | 0.062 | 0.015 | (0.039–0.098) | 0.062 | 0.015 | (0.039–0.099) |
| Covariance $\sigma^2_{v0v1}$ | | | | | | | -0.091 | 0.071 | (-0.231–0.048) | -0.098 | 0.071 | (-0.238–0.041) |
| ICC | | | | | | | | | | | | |
| $rho_{background}$ | 0.054 | 0.024 | (0.022–0.127) | | | | 0.011 | 0.011 | (0.001–0.076) | 9.64e-33 | 5.26e-18 | |
| $rho_{individual \mid background}$ | 0.579 | 0.026 | (0.527–0.629) | | | | 0.558 | 0.043 | (0.472–0.64) | 0.556 | 0.043 | (0.47–0.063) |

Notes:

\*\*\* p < 0.001,

\*\* p < 0.01, and

\* p < 0.05.

and drinking (logit AOR: 1.344, 95% CI: 1.163–1.554; Mixed 1 AOR: 1.651, 95% CI: 1.288–2.117 and Mixed 2 AOR: 1.649, 95% CI: 1.286–2.114) have deteriorating effects on mental health. This study, however, did not find any significant association of being obese and mental health for the study cohort in all our models. In the case of demographic variables, the study found that women are twice as likely as men to have a mental disorder (logit AOR: 1.484, 95% CI: 1.286–1.712; Mixed 1 AOR: 2.063, 95% CI: 1.461–2.913 and Mixed 2 AOR: 2.021, 95% CI: 1.431–2.851). However, the 'rural resident' variable was found to be significant in only the logit estimate (AOR: 0.759, 95% CI: 0.593–0.97). In addition, the study found not significant association between the sample period (time variable) and mental disorder of the study cohort.

In our findings, individual-level fixed effects have much stronger impacts on mental health than family and maternal background characteristics. We found that only household income and parental living arrangement (whether participants did not have the opportunity to live with both biological parents) were significant. Individuals who grew up in a poor household have approximately 1.6 times more likely to have mental disorder compared to youth who grew up in an affluent family (logit AOR: 1.258, 95% CI: 1.05–1.506; Mixed 2 AOR: 1.572, 95% CI: 1.017–2.43). Similarly, individuals who did not grow up with both biological parents in their childhood have approximately 1.6 times more likely to have mental disorder compared to the youths who grew up with both parents (logit AOR: 1.183, 95% CI: 1.017–1.376; Mixed 2 AOR: 1.586, 95% CI: 1.097–2.294). However, in our study, both mother's education and occupational status were not significant in any model. In addition, the random variances of family and maternal background in multilevel models were much lower compared to the null model (Null $\sigma^2_{v0}$: 0.423, 95% CI: 0.166–1.08 and Mixed 1 $\sigma^2_{v0}$: 0.078, 95% CI: 0.01–0.608). Indeed, the background variance disappears if fixed effect background group characteristics are considered. Contrary to background random effects, individual level intercept variances are much larger (Null $\sigma^2_{u0}$: 4.101, 95% CI: 3.353–5.017, Mixed 1 $\sigma^2_{u0}$: 4.068 95% CI: 2.878–5.749 and Mixed 2 $\sigma^2_{u0}$: 4.116, 95% CI: 2.921–5.8). In summary, rather than the group level family and maternal backgrounds, the driving forces in mental health outcomes of the youths are the individual-level characteristics.

## Discussion

The present study aimed to investigate the influence of group level family and maternal background characteristics and individual level circumstances-effort characteristics on the performance of youth mental health over time in Australia. Past research amassed substantial evidence in linking maternal education and occupation, with child's health outcomes [6, 7, 9]. However, contrary to this, we did not find any evidence linking youths' mental health with mother's education in any of our regression results. Perhaps, the thesis examined by Patrick West in earlier research plays a role in this context [26]. West argued that youth, in contrast to childhood, possess a process of equalisation which removes the influences of certain dimensions of family background differences (such as maternal education in our case) in youth mental health outcomes. Few studies have explored this area, and further work is needed for the youth age groups. It is possible that as youth become more independent that the influence of mothers' education becomes less important. We did, however, find significant impact of household income and family living arrangement on mental health performance of the youth. This impact is supported by other empirical literature [6, 7, 9, 27].

In order to investigate the underlying value judgement of individual effects, we followed equality of opportunity theory and categorised our variables into circumstances and effort groups [21, 22]. Our estimated results are consistent with the theory. We found that financial

shocks, life event shocks and long-term health conditions significantly deteriorate youth mental health condition. These findings are consistent with the adverse event literature [12, 28, 29]. In addition, we found that negative health habits such as smoking and drinking worsen mental health where as positive social habits such a club or sporting activities favours mental health, which is also in line with existing research [30]. Certainly, as youth become independent, the role of social relationships with those outside of families become particularly important in bolstering mental health.

One of the major contributions of this study is that we considered individual and group level variability through a multilevel modelling technique that other studies in the literature ignore. We found that there exists significant variability in individual level characteristics. In addition, individual level slope and intercepts also varied across time. However, compared with individual effects, the group level impact of family and maternal background characteristics did not vary. The implication of our finding is that, even though, some background dimensions (i.e., household income and living arrangements) have significant influences, the impact of maternal background is much smaller than the individual effects such as financial and adverse life events, long-term health conditions, and health behaviour related activities (smoking and drinking habits).

Our results and findings have some interesting implications. Our findings stimulate discussion about the mechanism of maternal background linking the mental health childhood and adult cohorts. The findings suggest, more research is needed both in childhood and adult cohorts to further our understanding as to the impact of maternal background. Whilst maternal background may shape health in early childhood, its role in shaping youth health and mental health may not be so clear. On the other hand, there are number of factors that are clearly linked to youth mental health trajectories, including their physical health during ages 15–19. For example, smoking and drinking have clearly negative consequences on youth mental health, whereas club activities have positive effects. Policy makers might therefore be interested in implementing health related behavioural interventions to promote both physical and mental health. Another observation of this study also suggests the importance of providing ongoing support to youth who have experienced financial and adverse life events in order to prevent long-term mental illness. This may include financial, care coordination and emotional support to manage the consequences of the adverse events in the short-term and trauma-informed psychological care in the long-term. Detailed research in the methodology and design of such interventions as well as estimation of the associated delivery costs of such program are needed.

A few limitations rise when interpreting the findings of this study. First, the primary outcome of this study is a dichotomized variable that serves as a proxy measure to identify people with mental disorder. This procedure overlooks the fact that an increase in the K10 score indicates an increase in suffering and an increased risk of serious mental problems. As a result, mental illness severity categories were excluded from our study. Second, we have used K10, as an instrument of psychological distress to measure the likelihood of having a mental disorder. Although this measure has the strength of reducing the self-reported bias of patient reported mental health outcome, it is still an instrument that primarily used to measure psychological distresses. Thus, a bit of caution is necessary when interpreting outcomes with such instruments. Future research is necessary to further explore our research questions with other valid instruments. Lastly, we do not completely rule out the potential of ecological fallacy in relation to maternal background variation. Given the paucity of evidence about the influence of maternal background variables in our study, caution is advised when interpreting this finding. Additional research is required to validate this issue.

## Conclusions

In summary, our findings contribute to current knowledge by drawing attention to the lack of impact of maternal background on youth mental health. Our study findings suggest that the influence of maternal background is significantly less than the individual impacts of adverse life events, chronic health disorders, and health behaviour-related activities. We also extend the scope of our research by using improved modelling techniques, for example, utilising multi-level modelling to assess mental health outcomes, which is another major contribution of this study.

It is imperative that future research examines further the link of maternal background between younger and older age cohorts. The main strength of our study is the use of an equality of opportunity framework and multilevel modelling techniques to address critical questions on youth mental health in Australia. Policy-wise, mental health interventions should consider heterogeneity of adverse youth circumstances and health-related behaviours. This research will provide essential insights into how to improve such interventions.

## Supporting information

**S1 Appendix.**
(DOCX)

## Author Contributions

**Conceptualization:** Rubayyat Hashmi.

**Data curation:** Rubayyat Hashmi.

**Formal analysis:** Rubayyat Hashmi.

**Investigation:** Rubayyat Hashmi.

**Methodology:** Rubayyat Hashmi.

**Project administration:** Rubayyat Hashmi.

**Resources:** Rubayyat Hashmi.

**Software:** Rubayyat Hashmi.

**Supervision:** Khorshed Alam, Jeff Gow, Sonja March.

**Validation:** Rubayyat Hashmi.

**Visualization:** Rubayyat Hashmi.

**Writing – original draft:** Rubayyat Hashmi.

**Writing – review & editing:** Rubayyat Hashmi, Khorshed Alam, Jeff Gow, Sonja March.

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
