## [Decision Letter · Decision Letter 0]

16 Jun 2021

PONE-D-21-14569

Does maternal background matter? A multilevel approach to modelling mental health status of Australian youth using longitudinal data

PLOS ONE

Dear Dr. Hashmi,

Thank you for submitting your manuscript to PLOS ONE. After careful consideration, we feel that it has merit but does not fully meet PLOS ONE’s publication criteria as it currently stands. Therefore, we invite you to submit a revised version of the manuscript that addresses the points raised during the review process.

The revised version should address all comments.

We look forward to receiving your revised manuscript.

Kind regards,

Petri Böckerman

Academic Editor

PLOS ONE

Journal Requirements:

"NO - The funders had no role in study design, data collection and analysis, decision to publish, or preparation of the manuscript."

Reviewers' comments:

Reviewer's Responses to Questions

**Comments to the Author**

1. Is the manuscript technically sound, and do the data support the conclusions?

Reviewer #1: Partly

Reviewer #2: Yes

2. Has the statistical analysis been performed appropriately and rigorously? 

Reviewer #1: I Don't Know

Reviewer #2: I Don't Know

3. Have the authors made all data underlying the findings in their manuscript fully available?

Reviewer #1: Yes

Reviewer #2: No

4. Is the manuscript presented in an intelligible fashion and written in standard English?

Reviewer #1: Yes

Reviewer #2: Yes

5. Review Comments to the Author

Reviewer #1: More references to be added to support the introduction, purpose and discussions. Additional comments are attached in the reviewed manuscript. Conclusion should be more specific and can't be generalized.

Reviewer #2: Mental illness is a leading cause of disability worldwide among young people (aged 10–24 years). Whilst there is a good body of work that has generated substantial evidence about the influence of maternal circumstances on child and adolescent mental health, a key methodological issue in most of them being their exclusive reliance on cross-sectional designs. Longitudinal analyses using repeated measures offer a richer understanding of the natural history as one progress in the life trajectory and can identify risk factors associated with transitions. In this context, the present manuscript on youth mental health using longitudinal data has potential scope for adding further evidence to the existing literature on child and adolescent mental health and will contribute towards life course perspective on mental health problems.

However, for better understanding and clarity, I have the following concerns to be addressed:

English language and grammar used throughout the manuscript needs thorough revision and editing

Purpose & Abstract: The method in the Abstract section should go beyond the description of data and provide the method (model, Analysis) used for answering the research question.

Keywords can be better refined to match the appropriate controlled vocabulary

Wherever possible, a better technical term should be used. For example, “ great emphasize or great importance” instead of “great store”; “important determinant/risk factor” instead of socioeconomic status (SES) is an “important marker.”

Introduction: As the authors have stated that “circumstances experienced by individuals in their childhood and adolescent period are certainly much different than the period when they are transitioning to youth and adulthood” it is important that how they have addressed or parametrized the time-varying covariates in their model. Currently, the manuscript falls short of providing the detail.

Also, it would be added value to show how the covariates have varied over time to substantiate the above statement.

In the term “to investigate the impact of youth circumstances”, circumstances here mislead one to assume those recent or current occurrences around youth period. They may not be appropriate to describe the maternal circumstances. The authors need to use better term to denote the maternal circumstances.

The specific objective or research question, or hypothesis must be mentioned explicitly besides the primary goal. Otherwise, it will allow readers to make their own assumptions.

Methods: The authors should have clarity and consistency in using the terms mental health problems, mental health distress and mental disorders. Mental disorders usually require a diagnostic tool. K10 is used to detect non-specific psychological distress, not mental disorders.

The authors do not specify the point in time at which the covariates for maternal background status are considered. Are these the data collected at baseline? If so, how would they take into account of time-varying components? Household income might change over time.

For other covariates, are baseline values used? Time-varying values? Values from the previous period?

For demographic covariates, there are many families that move from rural to urban, and some vice versa. How this was addressed in the analyses.

Authors need to provide brief details on how they have addressed repeated measures, multi-collinearity and interaction -effects in their model.

Probably, the author can bring more clarity to the nested structure through diagrammatic illustration. Further, instead of “maternal background history” the author should use “maternal background history groups”, as it allows one to assume it as an individual characteristic without the qualifier.

Results: The results section appears a little terse, and the language used to describe the results needs to be reconciled for better clarity. For example, one loses clarity on the sentence “All groups have approximately six to seven points of standard deviation which indicates considerable variability at the individual level”; it was unclear what the authors meant by “all groups” and how group value indicates individual variability. This information can be added in Table 1 for better clarity.

The authors can prefer to use the term “substantially” instead of significantly in the sentence “It can be seen that age groups do not vary significantly in mean K10 score both”, as it was not finding from statistical analyses.

Though not a statistical expert, it was not clear either from their description of their statistical analyses or from table 3 how they arrived at individual estimates for background characteristics if they were grouped to form level 3 with 16 groups. Suppose the background characteristic were entered at the individual level (2), then how they can be a component of level 3. For example, if maternal education is one component that contributed to form level 3 group, then how one can derive estimates for maternal education. The authors need to specify which variables corresponded to random slope and random intercept.

The components like household income, family living arrangement, maternal education and occupational status are very much a characteristic at individual level. However, the authors have used these components and formed 16 subjective groups. In that sense, I presume that there was no natural nesting. The authors need to provide strong rational for artificially creating a group and its practical relevance.

Discussion: Overall, the discussion needs strengthening by relating the results with the available literature and include strengths and limitations.

The authors need to discuss the limitation of dichotomizing the categorical outcome especially in the context that increase in K10 score signifies increase in distress.

The way the group (level 3) was conceptualized has a significant impact on the study finding; there are evidences linking maternal characteristics with some individual characteristics which were significantly associated with mental health of young. Without looking at moderation/interaction effects, any conclusion on maternal characteristic would be unconvincing.

6. PLOS authors have the option to publish the peer review history of their article (what does this mean?). If published, this will include your full peer review and any attached files.

Reviewer #1: **Yes: **Racha Abi Hana

Reviewer #2: No

---

## [Author Response · Author response to Decision Letter 0]

28 Dec 2021

Please check the response to the reviewers document for details.

---

## [Decision Letter · Decision Letter 1]

25 Jan 2022

PONE-D-21-14569R1Does maternal background matter? A multilevel approach to modelling mental health status of Australian youth using longitudinal dataPLOS ONE

Dear Dr. Hashmi,

Thank you for submitting your manuscript to PLOS ONE. After careful consideration, we feel that it has merit but does not fully meet PLOS ONE’s publication criteria as it currently stands. Therefore, we invite you to submit a revised version of the manuscript that addresses the points raised during the review process. The revised version should address the remaining comments.

We look forward to receiving your revised manuscript.

Kind regards,

Petri Böckerman

Academic Editor

PLOS ONE

Reviewers' comments:

Reviewer's Responses to Questions

**Comments to the Author**

1. If the authors have adequately addressed your comments raised in a previous round of review and you feel that this manuscript is now acceptable for publication, you may indicate that here to bypass the “Comments to the Author” section, enter your conflict of interest statement in the “Confidential to Editor” section, and submit your "Accept" recommendation.

Reviewer #2: (No Response)

2. Is the manuscript technically sound, and do the data support the conclusions?

Reviewer #2: Partly

3. Has the statistical analysis been performed appropriately and rigorously? 

Reviewer #2: I Don't Know

4. Have the authors made all data underlying the findings in their manuscript fully available?

Reviewer #2: No

5. Is the manuscript presented in an intelligible fashion and written in standard English?

Reviewer #2: Yes

6. Review Comments to the Author

Reviewer #2: Overall, the authors have addressed most of the concerns. However, based on the response by the authors, I have the following concerns:

The authors should remove the term “group” from “maternal background group” in all the places except in the statistical analyses section. It would not be appropriate and easily comprehensible to use the term “maternal background group” in all the places.

As the grouping involves HH income and family living arrangement, which are more of a family background characteristic, it would be misleading and erroneous to include and label it as maternal background characteristics. The face validity of the construct of maternal background becomes debatable when HH income and family living arrangement, which are more of a family background characteristic, were included to define the maternal background characteristics. The authors need to use a more valid and appropriate term as it has implications for the conclusions. This is particularly important when this paper makes a strong claims and conclusions about the lack of impact of maternal background on youth mental health.

In this context, the authors need to refine the title to reflect the content and context more accurately.

From the author's response, it became evident that the grouping was forced artificially to answer the research question, and the model did not follow natural nesting. The very nature of grouping and the subsequent modelling and conclusions does not rule out the possibility of ecological fallacy completely. This will be a serious concern, and the authors need to be cautious in making strong conclusions on the lack of impact of maternal background on youth mental health. Accordingly, the abstract and discussion section should be refined and revised further to account for the limitations of the grouping and modelling.

7. PLOS authors have the option to publish the peer review history of their article (what does this mean?). If published, this will include your full peer review and any attached files.

Reviewer #2: No

---

## [Author Response · Author response to Decision Letter 1]

1 Apr 2022

All comments are addressed. Please see the response to reviewers comments file for details.

---

## [Decision Letter · Decision Letter 2]

5 Apr 2022

Do family and maternal background matter? A multilevel approach to modelling mental health status of Australian youth using longitudinal data

PONE-D-21-14569R2

Dear Dr. Hashmi,

We’re pleased to inform you that your manuscript has been judged scientifically suitable for publication and will be formally accepted for publication once it meets all outstanding technical requirements.

Kind regards,

Petri Böckerman

Academic Editor

PLOS ONE

Additional Editor Comments (optional):

Reviewers' comments:

Reviewer's Responses to Questions

**Comments to the Author**

1. If the authors have adequately addressed your comments raised in a previous round of review and you feel that this manuscript is now acceptable for publication, you may indicate that here to bypass the “Comments to the Author” section, enter your conflict of interest statement in the “Confidential to Editor” section, and submit your "Accept" recommendation.

Reviewer #2: All comments have been addressed

2. Is the manuscript technically sound, and do the data support the conclusions?

Reviewer #2: Yes

3. Has the statistical analysis been performed appropriately and rigorously? 

Reviewer #2: I Don't Know

4. Have the authors made all data underlying the findings in their manuscript fully available?

Reviewer #2: No

5. Is the manuscript presented in an intelligible fashion and written in standard English?

Reviewer #2: Yes

6. Review Comments to the Author

Reviewer #2: I thank the authors for their careful responses and revisions to their paper in response to my comments

7. PLOS authors have the option to publish the peer review history of their article (what does this mean?). If published, this will include your full peer review and any attached files.

Reviewer #2: **Yes: **Senthil Amudhan

---

## [Editor Report · Acceptance letter]

13 Apr 2022

PONE-D-21-14569R2 

Do family and maternal background matter? A multilevel approach to modelling mental health status of Australian youth using longitudinal data 

Dear Dr. Hashmi:

I'm pleased to inform you that your manuscript has been deemed suitable for publication in PLOS ONE. Congratulations! Your manuscript is now with our production department. 

Kind regards, 

on behalf of

Professor Petri Böckerman 

Academic Editor

PLOS ONE